# Percept Activation Graph (PAG): Decomposing LLM Computation into Perceptual Entities and Their Interactions

## Abstract

Understanding the computations performed by large-scale neural network models remains an important challenge. Recent work has motivated holistic approaches that focus on population-level dynamics of neurons in these networks, suggesting that these dynamics reflect statistical regularities in data and that the human perceptual tendency for chunking can be leveraged to identify recurring cognitive entities. We extend this line of work by introducing new techniques, inspired by cognitive science and neuroscience, to analyze LLM computations. We formalize chunking in neural data through the perceiving function, which maps recurring high-dimensional activities into a dictionary of recognizable entities. Building on this definition, we decompose neural activations in large-scale networks into a finite set of chunks and find that model activations exhibit compressible regularities across both tokens and layers. Based on these chunks, we define the Percept Activation Graph (PAG) which captures the causal structure of chunks across layers. We apply this analysis to LLMs to examine how they represent compositionality in context, analyzing layer-wise activations during in-context learning on the SCAN meta-learning dataset. Within the PAG, we identify distinct components that encode primitives, and demonstrate that perturbing these components predictably alters the model's compositional generalization behavior. Our method provides a pathway to automatically extract structured relations between chunks that causally and controllably influence the computation of large-scale neural networks.

## 1 Introduction

Remarkable progress has been made in scaling architectures and improving performance in large-scale neural network models such as large language models (LLMs). However, the internal computational mechanisms that give rise to their intelligent behavior are still poorly understood. As activations are high dimensional in nature, analyses at the level of individual neurons offers limited insight (Radford et al., 2017; Elhage et al., 2022; Wang et al., 2022b; Dai et al., 2022).

This difficulty has motivated more holistic approaches that focus on neural population-level dynamics. Recent work (Wu et al., 2025a) hypothesized that such dynamics reflect statistical regularities in the data, and that the human perceptual tendency of *chunking* can be leveraged to identify neural activities reflective of recurring concepts or patterns in the data.

In this paper, we build on top of this work and lift the reflective restrictions by introducing new cognitively inspired conceptualization and techniques. I.e. there may very well be population-level activities that the network create on its own, and may not correspond to easily identifiable patterns in the training data, but are nontheless interpretable. This is achievable by leveraging the property in cognition that chunks high dimensional visual data into a set of entities. We formalize this process as the *perceiving function*, which takes high-dimensional data as input and outputs a dictionary of recurring patterns within. Applying this perceiving function on neural data extract recurring population-level states that compresses network activations, and suggest regularities in processing states both across time and successive layers.

Based on these extracted entities, we define the Percept Activation Graph (PAG), which captures the causal interactions of how state propagates across layers to summarize the computation of a

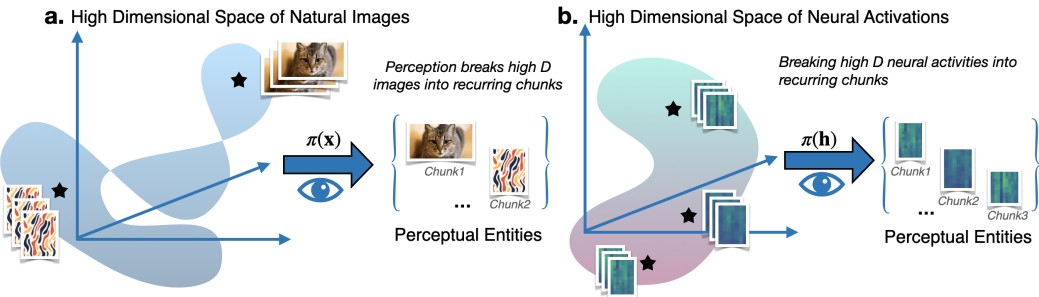

Figure 1: Human vision naturally segments the high-dimensional image manifold by learning a dictionary of recurring entities (chunks). Each entity corresponds to a revisited location, or cluster, within sub-dimensions of this manifold. Some entities can be associated with linguistic labels, while others remain pre-linguistic. We exploit this same property to identify entities within high-dimensional neural activity in artificial neural networks and apply unsupervised methods to extract recurring patterns in LLM's model activations.

network processing sentences from a corpus. We then apply this technique to interpret the interal activiations of LLM as it learns compositional generalization in context. From a subset of the SCAN meta-learning task, we extract PAG and show the LLM's internal state processing and propagation as it parses subsequent tokens in-contenxt. Within the PAG, we identify distinct components that encode primitives and modifiers, and we show that swapping the primitives predictably induces hallucination of the network to apply the intended false primitives to the modifier.

In sum, we propose techniques inspired by cognitive neuroscience to interpret computations in large-scale networks, and make the following contributions:

- We provide a precise definition of *perceiving function*. This characterizes the cognitive process that decomposes the high-dimensional perception into a dictionary of recognizable entities.
- We show that applying the perceiving function on network activations reveals temporal motifs and compressible neural activation patterns.
- We introduce the Percept Activation Graph (PAG). This introduces a major difference and advancement from the prior view and formulates perception and interpretable entities as a graph.
- We show that the connected components of this graph are causally connected to each other in a highly complex nature language corpus.
- We apply this method to analyze network activations during in-context learning on a subset of the SCAN dataset. Within the PAG, we identify distinct components for primitives and modifiers, and show that perturbing these components predictably alters the model's compositional generalization behavior.

Our approach is the first to extract interpretable, structured components of network computation directly from neural embeddings to the simplicity of a finite state machine defined by PAG, and paves the way for future methods to identify computational states driving the behavior of LM of any kind.

## 2    RELATED WORK

A Large Language Model (LLM) receives a prompt and produces text in response. We want to understand how internal computations give rise to the behavior that we observe.

Understanding is the process by which perceptual input gives rise to cognitive entities, which are then structured into relational frameworks that enable the prediction of observations (Miller, 1956; James, 1890). Human understanding and learning are progressive in nature. We tend to identify

recurring patterns in our experience and abstract these as distinct mental entities. These entities are then understood in terms of their relationships to one another. Over time, previously acquired concepts serve as foundational building blocks for more complex, sometimes abstract, structures of knowledge (Piaget, 1952; Wu et al., 2022; 2025b).

The entities that constitute our understanding differ quite a bit in approach to understanding artificial neural networks. Initially, researchers sought to study the activity of individual neurons and their relationship to a neural network's inputs (Geva et al., 2022; Zou et al., 2023; Belinkov, 2022; Belrose et al., 2023; Pal et al., 2023; Din et al., 2023). However, only a small fraction of neurons has their activation correlated with semantically meaningful concepts in the input, and the activity of any single neuron is rarely sufficient to alter network behavior (Radford et al., 2017; Elhage et al., 2022; Wang et al., 2022a; Dai et al., 2022; Voita et al., 2023; Miller & Neo, 2023), while most neurons in artificial neural networks exhibit activities associated with multiple often unrelated concepts (Mu & Andreas, 2021; Elhage et al., 2022; Olah et al., 2020; Gurnee et al., 2023). It remains unclear how the activity of a vast number of *polysemantic* neurons contributes to a model's internal computations and, consequentially, its behavior.

One step further from this approach is to use, instead of neurons in the network as an entity for understanding, neurons in the hidden layer of sparse autoencoder. By projecting neural activities into a higher-dimensional latent space — i.e., using more hidden units than input dimensions — and enforcing sparsity constraints, the model encourages the emergence of monosemantic hidden units in the latent space, whose activity correlates with a distinct concept in the input (Gao et al., 2024; Makhzani & Frey, 2013; Cunningham et al., 2023; Chaudhary & Geiger, 2024; Karvonen et al., 2024). However, interpreting the hidden units of an SAE remains a significant challenge. While prior work has highlighted some interpretable features - such as units that activate when seeing the phrase "Golden Gate Bridge" across different languages - SAE units are continuous in their activity and do not correspond to a discrete cognitive entity. To perturb SAE units, their activations are typically scaled by a chosen factor, but the choice of this factor is ad hoc and lacks consensus within the community. Moreover, the causal role of SAE units is often ambiguous, influencing network computation only part of the time (Wu et al., 2025c).

Another way of interpreting network activations is to study the low-dimensional projection of neural activities. Representation engineering methods, for example, project internal activations while the network processes text into a low-dimensional principal component (PC) space. By identifying concept vectors within this space, researchers can steer the network's internal representations in alternative directions (Zou et al., 2023). However, this approach is constrained by the granularity of the concepts: it works reasonably well for coarse-grained properties such as bias, but tends to fail when applied to fine-grained semantic distinctions.

This highlights the need for more holistic approaches that account for the distributed nature of neural representations. Recent work Wu et al. (2025a) posits the *reflection hypothesis*, which demonstrates distributed neural activities exhibit patterns that reflect regularities in the data. And these platonic reflections can be extracted as interpretable cognitive chunks chunks. While useful, assuming chunks are reflections of external input limits interpretability strictly to pre-existing regularities in data. This limitation motivates our paper to formalize the definition of the perceiving function and apply it to segregate emergent cognitive entities from neural activations.

## 3 PERCEIVING FUNCTION: BREAKING HIGH DIMENSIONAL NEURAL MANIFOLD INTO CONCRETE PERCEPTUAL ENTITIES

From millions of visual pixels each moment, we effortlessly perceive coherent objects. Cognition accomplishes this by chunking sensory input into discrete entities that segment and predict observations, transforming high-dimensional data into structured representations (Graybiel, 1998; Gobet et al., 2001; Egan & Schwartz, 1979; Ellis, 1996; Koch & Hoffmann, 2000; Chase & Simon, 1973; Wu et al., 2023; 2025b). This universal tendency to form chunks allows continuous perceptual streams to be reduced to a finite set of recurring, interpretable units. Geometrically, a sequence of sensory inputs traces a trajectory through a high-dimensional pixel space. When an image recurs, this trajectory revisits a similar region of the manifold. For example, alternating between two images—a cat and an amorphous pattern—produces a path that oscillates between two distinct points in pixel space. Cognition compresses this continuous trajectory into a sequence of perceptual enti-

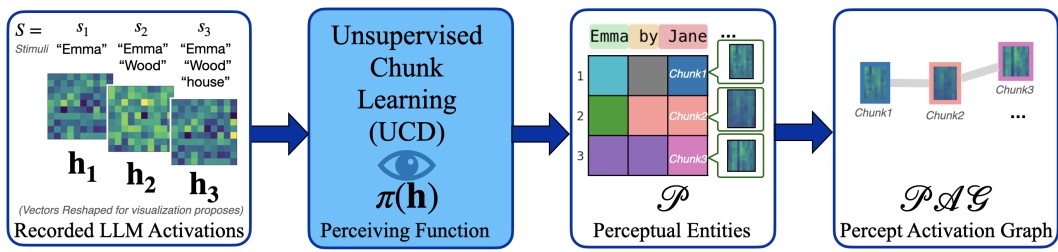

Figure 2: Extracting a Percept Activation Graph from LLM Activations

ties, "cat" and "some pattern," and thereby transforming high-dimensional variation into a sequence of meaningful symbols. These entities need not always be linguistic: some correspond to words, whereas others remain sub-symbolic or pre-linguistic (e.g., a familiar yet unnamed shape; as in Figure 1). Recognizing these pre-linguistic entities are the basis for learning specific names or labels for these entities e.g. "shape xyz" (Barsalou, 1999; Rosch et al., 1976).

**Defining the Perceiving function $\pi(\mathbf{x})$ that breaks high dimensional data into chunks** We can formalize *chunking* in vision as a perceiving function $\pi(\mathbf{x}) : \mathbb{R}^d \to \mathcal{P}$, that maps an image $\mathbf{x}$ in the manifold of natural images $\mathcal{H} \subset \mathbb{R}^d$ to a set of a cognitive entities $\mathcal{P}$, which contains a set of chunks (e.g., all images perceived as the same "cat" / the same amorphous shape). Following the convention of (Wu et al., 2025a), a *chunk* as a subspace of $\mathbb{R}^d$ that are equivalently identified by perception as the same entity. For example, $c_{\texttt{cat}}$ and $c_{\texttt{dog}}$ occupies distinct subspaces in $\mathbb{R}^d$. By definition, chunks are interpretable: they correspond to the kinds of recurring perceptual entities that structure our visual and conceptual experience. Classic theories in perceptual organization suggest structured units of perception arise from experience. These entities become the basic units through which we segment and understand perceptual input (Rosch et al., 1976; Schyns et al., 1998), and emerge prior to linguistic labeling.

Similar to natural images, neural population dynamics is known to lie on a low-dimensional manifold. We apply the perceiving function to artificial neural activities, asking whether high-dimensional activations can be decomposed into recurring entities ;just as vision compresses sensory input into a set of chunks, a perceiving function $\pi$ can segment LLM activations into a finite set of frequently revisited subspaces. These segments constitute "chunks"—discrete neural patterns that correspond to recurring computational states in the manifold.

Formally, The activations of an artificial neural network $\mathbf{h}$ lies in a high-dimensional manifold $\mathcal{M} \subset \mathbb{R}^d$, Applying the same **perceiving function** on hidden state vectors $\mathbf{h}$ from the manifold $\mathcal{M}$ would give: $\pi(\mathbf{h}) : \mathcal{M} \to \mathcal{P}$, where $\mathcal{P}$ is a set of perceptual chunks. Each element $e \in \mathcal{D}$ corresponds to a chunk label. Each point $\mathbf{h}_i \in \mathcal{M}$ denotes the activation of the network processing sequence up until a particular token index $i$. We can then define a **Chunk** $c_e \subset \mathcal{M}$ associated with perceptual entity $e \in \mathcal{D}$ as: $c_e = \{\mathbf{h} \in \mathcal{M} \mid \pi(\mathbf{h}) = e\}$ That is, $c_e$ is the region of the manifold consisting of all neural population vectors that are identified by $\pi$ as the same perceptual entity $e$.

Let $\mathcal{S}$ be a sequence of stimuli $(s_1, s_2, \cdots)$, e.g. a successive input fed into the network. The LLM maps each stimulus $s \in \mathcal{S}$ to a sequence of hidden states $(\mathbf{h}(s_1), \mathbf{h}(s_2), \cdots)$ (Figure 2). For a stimulus $s$, the set of percepts $\mathcal{P}$ elicited by $s$ can be defined as the set of chunks that $\pi$ maps to $\mathcal{P}(s)$. For a sequence of stimuli $\mathcal{S}$, we can then define a percept set induced by $\mathcal{S}$ as $\mathcal{P}(S) = \bigcup_{s \in S} \mathcal{P}(s)$.

Much like abstract concepts emerge from concrete concepts in human cognitionMurphy (2002); Wu et al. (2025b), and abstract concepts understood through systems of relations—taxonomic, functional, and compositional (Barsalou, 1999; Lake et al., 2017). We formalize a relation structure between chunks as the **Percept Activation Graph (PAG).** PAG is a directed weighted graph $PAG = (V, E, w)$ defined on the percept $\mathcal{P}(S)$ as the network processes stimuli $S$. The percept contains a set of chunks $\mathcal{P} = \{c_1^1, c_2^1, c_1^2, c_2^2 \cdots\}$ from each layer of the network. $E \subseteq V \times V$ is the set of directed edges, where an edge $(c_i^l, c_j^{l+1}) \in E$ exists if chunk $c_i^l$ in layer $l$ leads to chunk $c_j^{(l+1)}$ in the subsequent layer $E = \{(c_i^l, c_j^{l+1}) : p(c_i^{l+1} \mid c_j^l) > 0\}$. The edge is weighted by

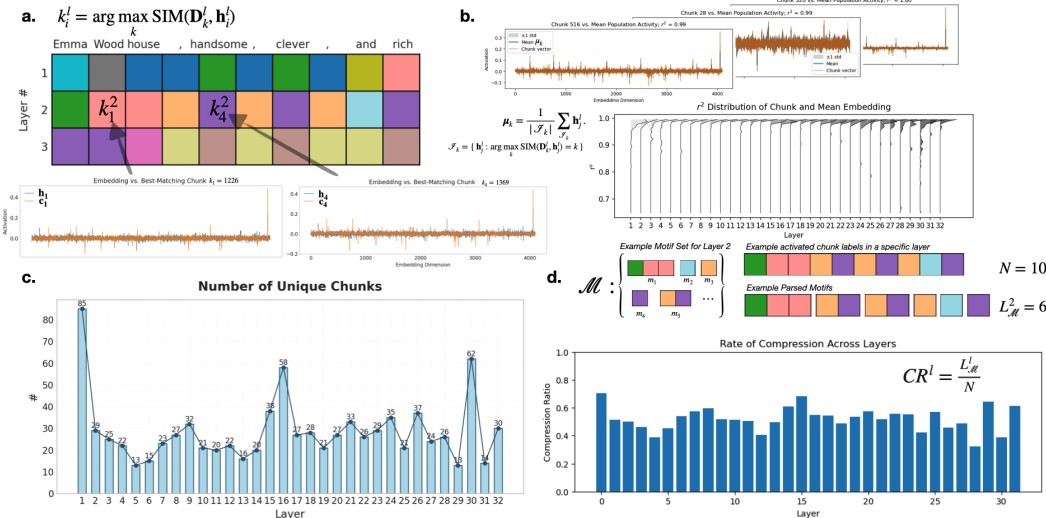

Figure 3: Applying the perceptual function to LLaMA-3 activations on sentence-wise prompts from the *Emma* corpus. a. Above: Tagging observations with chunk identities provides a compressed representation of neural computation across layers. Below: several embeddings and the chunks tagging the embeddings. b. Above: several example chunks, and the mean embeddings tagged by the chunks. Below: Distribution of $r^2$ values between each chunk and the mean embeddings of the activations it tags, evaluated across all layers. c. Size of percepts across layers. d. Above: Example chunk activation motifs across tokens. Below: Compression ratio across all layers of the network evaluated on the corpus

$w(c^i, c^j) = p(c_i^{l+1} \mid c_j^l) = \frac{N(c^i, c^{i+1})}{N(c^i)}$. $N(c^i, c^{i+1})$. The empirical estimation of activation probability for chunk $c_i^{l+1}$ at layer $l+1$ given the activation of chunk $c_j^l$ at layer $l$. The PAG encodes the probabilistic transition structure of chunk across layers, describing the population-level computation in the network as a finite state machine. We use of LLaMA-3-8B (Dubey et al., 2024) as the sample LLM under study, chosen for its open-source availability and architectural complexity.

## 4 Layer-wise Outputs of LLaMA-3 on Natural Language Prompts Exhibit Structured Regularities and Compressible Motifs

**Extracting Percepts**   We analyze first the layer-wise output activation of LLaMA-3 incrementally processing prompt sampled from a literature corpus *Emma*. The corpus was accessed via NLTK (Bird et al., 2009) from the Project Gutenberg (Hart, 1971), and in total contains more than 200,000 tokens. We split the corpus into individual sentences as prompts. And collect a dataset of neural activities comprised of embeddings from every hidden state output of the forward pass, as the model incrementally processes each newly appended tokens in each prompt. For a prompt with $m$ tokens, we will get an $m$ activation vectors $\mathbf{x}^l$ of dimension $d$ at layer $l$. Thereby, for each layer of LLaMA-3, we get a dataset of dimension 4096 times 200,000+ in total consistent of prompt from the corpus.

The question of learning a set of percepts can then formulated as learning a dictionary matrix $\mathbf{D}^l \in \mathbb{R}^{K \times d}$ that extracts the recurring patterns in the latent space of dimension $d$ for the layer $l$. We adopted the unsupervised chunking learning algorithm (UCD) that was originally intended to extract patterns reflecting of data regularities (Wu et al., 2025a) to this problem. UCD learns a dictionary of patterns $\mathbf{D}$ from batched data $\mathbf{X}$ of size $m$ via optimizing the loss function $\mathcal{L}(\mathbf{X}, \mathbf{D}) = -\frac{1}{M} \sum_{m=1}^{M} \max_{k \in \{1, \ldots, K\}} \mathrm{SIM}(\mathbf{D}_k, \mathbf{X}_n)$, which encourages each activation $\mathbf{X}_m$ in the batch to match its most similar dictionary column $\mathbf{D}_k$ measured by normalized cosine similarity $\mathrm{SIM}(\mathbf{D}_k, \mathbf{X}_n) = \frac{\mathbf{D}_k^\top \mathbf{X}_n}{\|\mathbf{D}_k\|_2 \|\mathbf{X}_n\|_2}$. One dictionary $\mathbf{D}^l$ was trained for each layer on the activation data for 100 epochs with a batch size of $n = 32$, with dictionary size $K = 2000$, using ADAM optimizer with a learning rate of $1 \times 10^{-2}$.

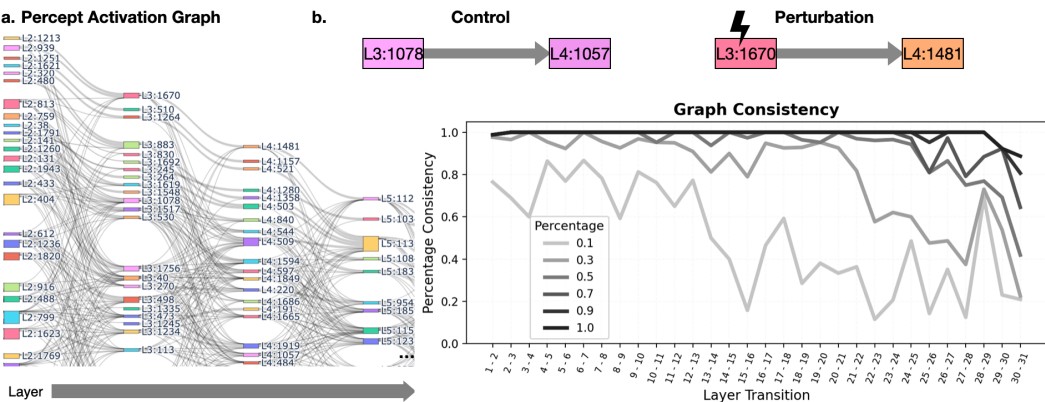

Figure 4: Percept activation graph (PAG). a. A part of the PAG extracted from the first all layers of LLaMA-3 while processing sentences from the Emma corpus, showing how chunk states transition across layers (full graph and conditional probabilities are shown in the supplementary HTML file.) b. Causal validation of PAG edges: grafting a chunk at layer $l$ and measuring the resulting chunk at layer $l + 1$ reveals that intervention-induced transitions closely follow the edges predicted by the PAG. This correspondence deteriorates when only random subspaces of the chunk vector are grafted.

These extracted chunks serve as the basic units for analyzing internal computation across layers. For each layer, we assign the label $k_i = \arg\max_k \mathrm{SIM}(\mathbf{D}_k^l, \mathbf{h})$, which tags the momentary embedding by the chunk which resembles it the most. Shown in Figure 3 a, this mapping reduces the 4096-dimensional activation at each layer to a single discrete chunk label, yielding a sequence of discrete chunk indices (visualized as colors) that compactly summarizes how the network processes the prompt across depth. This is analogous to how cognition parses high-dimensional perceptual data into concrete perceptual entities. Figure 3 b shows several example chunks, the average embeddings tagged by the chunk, and the standard deviation of each dimension. Across all layers, The learned chunks correlate strongly with the tagged embedding average. Additionally, the number of used chunks differ across layers, with the first and close to the last having the most.

We moved on to quantized the chunk alternation patterns by using a pattern learning algorithm (Wu et al., 2022) to extract a set of motifs $\mathcal{M}^l$ characterizing patterns of chunk labels across consecutive tokens. As each motif $m$ spans $|m|$ tokens, the length of the sequence parsed by cross-token motifs is described as $L_{\mathcal{M}} = \sum_{m \in \mathcal{M}} |m|$. The compression ratio is calculated as the compressed length divided by the original length $CR = \frac{L_{\mathcal{M}}}{N}$. $CR \leq 1$ suggest the existence of cross-token motifs. Figure 3 d shows the compression ratio across all layers of the network calculated from the entire corpus. A baseline ratio of 1 corresponds to the case where the network-elicited chunks are independent and exhibit no patterning. Across all layers of process, we observed a compression ratio below 1, which indicates substantial regularities and many recurring cross-token chunk patterns. This suggests the presence of computational motifs that span consecutive tokens of network processing.

Finally, 3 d shows the number of uniquely identified chunks across all layers in LLaMA-3 processing the corpus. We observe that early and late layers tend to have more number of unique chunks than the middle layer. This observation also align with other findings that early layers tend to compress information and later layers needs to decompress.

## 5 INTERPRETING LAYER-WISE COMPUTATION THROUGH THE PERCEPT ACTIVATION GRAPH

Human cognition forms abstract concepts by first distinguishing perceptual entities and then understanding how these entities relate to one another through structured systems of relations (Murphy, 2002; Barsalou, 1999). In an analogous way, the Percept Activation Graph (PAG) provides a relational formalism for neural network computation and describes how learned chunks contextualize and influence one another across layers.

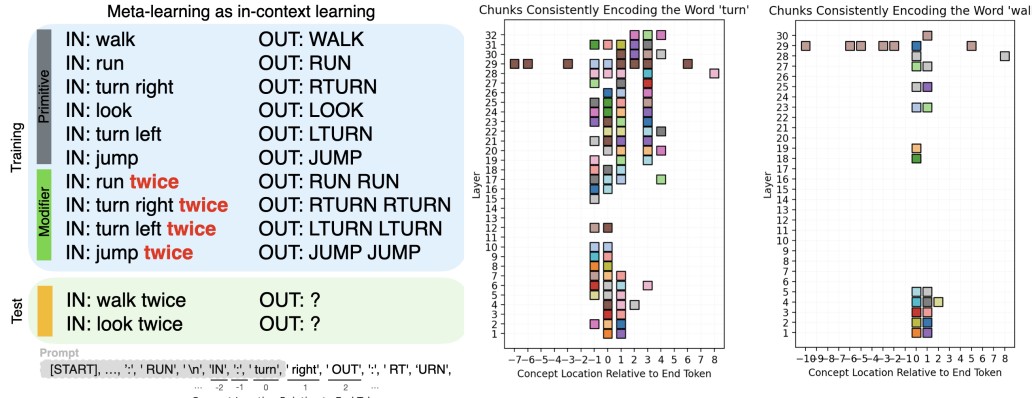

Figure 5: Measuring chunks that play roles for an in-context compositional generalization task. Left: We take a simplified version of the SCAN dataset and reformatted as an in-context meta-learning task, we give LLMs prompts containing the example input output pairs of all primitives and four modifier–primitive examples. Middle/Right: Chunks (across all 32 layers of LLaMA-3) that persistently activate when the concept occurs, relative to the end token of the prompt (+1 means the concept occurs subsequent to the prompt end (the network needs to predict the concept)).

After extracting the chunks, the perception is the set of activated chunks as the network processes prompts from the corpus. We then extract a percept activation graph, which describes the circuit of chunk interactions as the network processes the corpus. PAG contains nodes as activated chunk labels and edges specify the nonzero conditional probability of the subsequent-layer chunk given the chunk in the previous layer. Figure 4 a shows a subgraph of the PAG extracted from LLaMA-3 processing the Emma corpus (the full graph is provided in the supplementary HTML file).

A central question is whether these graph components reflect causal interactions rather than mere correlations. To test this, we intervene on the model's internal state by grafting a learned chunk into the population activity at a given layer and then measuring the chunk that emerges as most similar to the resulting hidden-state vector at the next layer.

Figure 4 b illustrates this procedure. Under natural activations, the model transitions from chunk 1078 at layer 3 to chunk 1057 at layer 4. When we instead graft chunk 1670 at layer 3, the model reliably transitions to chunk 1481 at layer 4—precisely the successor predicted by the PAG. Specifically, for each learned chunk in the PAG, we performed the following procedure: (1) replace the chunk into the population activity at a given layer; (2) measure which chunk in the subsequent layer dictionary $\mathbf{D}^l$ was maximally similar to the resulting hidden population activity; and (3) evaluate whether this consequentially emerged chunk was connected to the grafted chunk in the computational interaction graph. This process was repeated iteratively across all layers.

These experiments reveal that PAG edges capture robust causal dependencies: grafting a chunk at one layer almost always induces a connected chunk in the subsequent layer in PAG. The connected edges in PAG do not merely summarize statistical co-occurrence but carries the information about the causal transition inside a model's internal computation.

We additionally performed a set of control baselines by grafting partial chunk vectors—randomly subsampling 10%, 30%, 50%, 70%, or 90% of a chunk's dimensions—before repeating the intervention analysis. Figure 4 b shows that reducing the dimensionality of chunks substantially degrades consistency, especially near the output layer. These degradations confirm that the complete chunk representation is required to capture the true causal influence between layers. Overall, PAGs provide an interpretable abstraction of an LLM's computatio and connects neural-population dynamics to discrete, rule-like transitions to the simplicities of finite-state machines.

## 6 FINDING CHUNKS THAT DRIVES COMPOSITIONAL GENERALIZATION

As the PAG becomes complex on natural language corpus, we use a controlled task to test if the components of PAG can predictably alter model behavior, as it remains a mystery how large language models (LLMs) are able to infer and generalize rules from only a few in-context examples.

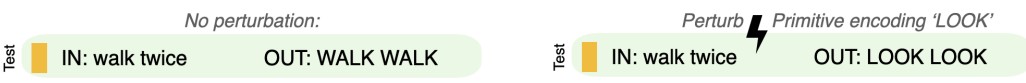

Figure 6: Illustration of primitive swapping in LLaMA-3: as the model generates hold-out prompts with unseen primitive–modifier combinations, replacing the chunks which encode primitive alters the output.

Equally unclear is how these rules, along with their constituent operations, are internally represented and applied. We show that by extracting a PAG, one can gain insight into the internal mechanisms that enable an LLM to generalize rules from limited in-context examples.

To this end, we use SCAN, a meta-learning benchmark with example sentences contains simple primitives and rules applied on these primitives (Lake & Baroni, 2023) organized in an input and output format. e.g. IN: jump, OUT: JUMP. We reformulate a subset of SCAN tasks as in-context meta-learning problems. As LLMs have demonstrated superior in-context learning abilities, we would like to use PAG to show the percepts that are responsible for compositional generalization in context.

In SCAN, each task consists of a command prefixed with IN: and its corresponding action sequence following OUT: (see Figure 5). There are 6 primitive actions (jump, walk, run, turn left, and turn right), and a number of rules. e.g. twice, thrice, etc. Without modifiers, primitive as an input needs to generate the capitalized varied form of primitive as the output (e.g., IN: jump → OUT: JUMP). The task of the network is to generate the correct action sequence after observing a number of command–action examples (e.g., IN: jump twice → OUT: JUMP JUMP).

To simplify the problem, we focus on a single modifier, twice, from SCAN modifiers. When applied to a primitive command, this modifier requires the corresponding action to be repeated. For example, the command IN: jump twice should be followed by the generated the action sequence OUT: JUMP JUMP.

We constructed 100 prompt samples by randomly ordering illustrations of all primitive commands, followed by four illustrations of modifier applied to four randomly selected primitives to control for the position effects. The remaining two primitives were reserved for the hold-out test prompt.

We recorded the forward-pass neural activity of LLaMA-3 as it incrementally processed tokens from the start of the sentence up to the ith token, reflecting the model's internal state as it predicts the i+1th token. We then applied the unsupervised chunk discovery algorithm on the activations.

**Chunk activations encode memory, reception, and prediction of words** We then allocated the chunk corresponding concept by finding the persistently activating chunks when the concept occurs relative to the end token. Two examples are shown in Figure 5. The network elicits distinct chunk activations depending on the position of "turn" within the sequence. When "turn" appears in the final, second-to-last, or third-to-last position, it reliably triggers different sets of chunks in a number of layers. Many commonly occurring chunks, active across multiple layers, encode the reception of "turn" as the latest token. When "turn" occurs earlier (particularly at an offset of –1, often before "right" when the model is preparing to generate the modifier for a primitive) the same chunks are also engaged. Interestingly, chunks encoding "turn" can still be observed at offsets of –6 and –7, suggesting that the concept continues to shape predictions well before the final token, reflecting a form of memory. Finally, a subset of chunks encode predictions. Some of these emerge when the network anticipates "turn" as the next token, while others predict its occurrence several steps ahead—appearing two, three, or even four tokens before "turn" is actually generated.

Similar analyses applied to other concepts reveal a persistent memory of "walk": when "walk" appears before the final token, it elicits chunks that continue to directly influence the network's output.

**Extracting a Circuit-level Diagram** We can then visualize the percept activation graph plotting the chunks in the percept $\mathcal{P}$ as the network processes the input: "IN: run twice OUT: RUN RUN ". Figure 7 shows the PAG, tracing the activated chunks through the layers, and observing how chunks in the previous layer influence chunks in the later layer. By visualizing conditional probabilities

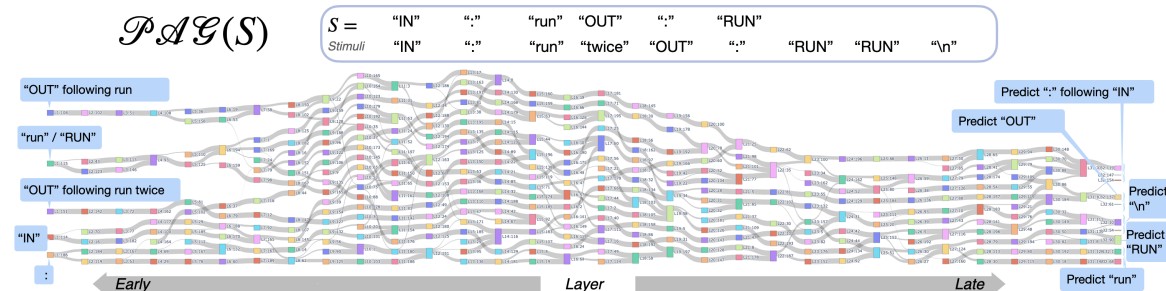

Figure 7: Percept activation graph for all run related tokens in the SCAN task. Chunks elicited by any tokens from IN: run OUT: RUN or IN: run twice OUT: RUN RUN, and the subsequent chunks activated inside the network. Edges link chunk activities between successive layers, with edge thickness indicating the magnitude of the transition probability between the connected chunks. Input chunks are associated with more textual meaning.

of chunk activations across layers, we observed cases where activity is effectively deterministic: a token triggers a stable chunk that reliably leads to another. The PAG reveals distinct neural states that depend on the input structure. For example, the population state following OUT after run diverges sharply from the population state following OUT after run twice. While the model initially encodes run and RUN in a shared state at early layers, these paths split in later layers as the network resolves the modifier twice. The graph also shows that the network constructs multiple distinct internal states to predict the same token. For instance, chunks 119, 147, and 154 at layer 32 all predict the colon following IN, and chunks 137 and 61 each independently predict OUT.

To test the causal roles of such chunks, we correlated the occurrence of each unique word appearing as the last token in the prompt with each of the chunk activities, and tagged chunks whose activity is strongly correlated with words ($r \geq 0.95$) as the word-encoding chunks. This allows us to find concept encoding chunks and test the causal role of replacing the activity of relevant layers which contain the concept-encoding chunks. As a demonstration, we specifically found the chunks separately encoding the primitives in the capitalized form of output: RUN, WALK, LOOK, LTURN, RTURN, JUMP, and tested whether we could swap the activations during test time. To do this we give the network a training prompt followed by the first part of the test prompt that contains one of the hold-out primitives applied to the twice modifier (IN: walk twice OUT:, as an example in Figure 5). Normally, the network generates WALK WALK in the following token. We then test the causal role of the target-encoding chunks by swapping the activation with the target-encoding chunks (such as jump) and measured whether the network generates the altered primitive applied to the modifier that we intend to target (such as the network outputs JUMP JUMP after OUT:). Table 1 shows the success rate of swapping the primitive which is applied to the modifier twice by replacing the layer activation with the target-encoding chunks. The success rate is calculated on iteratively grafting the primitives on all prompts in the dataset. Generally, except for RUN, we can swap the application of primitive successfully with the intervention, suggesting that the chunks extracted play a causal role and change the network behavior in a predictable way.

| Primitives Swap | Success Rate (%) |
| --- | --- |
| WALK | 87.5% |
| LOOK | 100% |
| RUN | 12.5% |
| JUMP | 79.2% |
| LTURN | 100% |
| RTURN | 87.5% |

Table 1: Success Rate for inducing controlled hallucination by finding chunks that accounts for primitives and swapping primitives to other network computational states that accounts for other primitives. Success rate is evaluated on All SCAN Prompts (with primitives presented in a random order) and Primitives

## 7 CONCLUSION

Understanding computation in large-scale neural networks remains an open challenge. We addressed this problem by formalizing chunking through the *perceiving function*, extracting discrete chunks that reveal compressible regularities in transformer activations. From these, we defined the Percept Activation Graph (PAG), which captures the causal structure of entity interactions across layers. Applying this framework to LLMs, we identified components for primitives and modifiers and showed that perturbing them predictably alters compositional generalization. Our method thus offers a principled approach to extracting interpretable PAGs that exert causal and controllable influence over large-scale models.

## 8 DISCUSSION

The key shift we propose is conceptual: the cognitive process of breaking perceptual data into chunks can be automated as an entity-extraction method that distills interpretable units directly from activity patterns. With minor adaptations, this framework extends naturally to other interpretability methods such as SAEs (Gao et al., 2024; Makhzani & Frey, 2013; Cunningham et al., 2023; Chaudhary & Geiger, 2024; Karvonen et al., 2024).

The method contains a light computational overhead. The chunk dictionary is learned offline, but The PAG can be constructed efficiently during inference. Grafting and interventions operate online as the model processes new inputs. In SCAN, we show that manipulating chunk activations in real time produces reliable, controlled hallucinations, demonstrating that PAG supports dynamic, behaviorally significant interventions.

That said, our approach has limitations. The sensitivity and granularity of extracted chunks, as well as the clarity of their interpretation, remain open to improvement. Future work should explore how computational entities support learning, and in particular, which entities contribute predictive signals that support memory, and how they interact. In this paper, we analyze embedding data from the layer output, which is subsequent to the attention processing in the network, we encourage future work to study the embedding data from the attention layers and other computational components of the network.

Additionally, chunks provide one useful granularity for decomposing the neural manifold, but they are not the only option. Alternatives include tracking the activation of sparse autoencoder (SAE) features or identifying fixed points, limit cycles, and attractor states by modeling the network as a dynamical system. All of these approaches aim to extract low-dimensional, discrete representations from high-dimensional neural activity. Our contribution is to show that such representations can be formalized as discrete, identifiable computational entities, offering a new framework for understanding network processes.

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

## ETHICS STATEMENT

This work investigates model activations by cognitive-inspired interpretability methods. Large Language Models (LLMs) were used only to aid and polish the writing; no ideas, analyses, or experiments were generated by them. We believe this disclosure is important for transparency.

## A   SUPPLEMENTARY INFORMATION

### A.1   CAUSALLY LINKED CHUNKS ARE NOT IDENTICAL

We conducted additional control experiments designed specifically to test whether the causally linked chunks are genuinely distinct or merely copies propagated through the residual stream.

The concern is, the causal experiment in Section 5 may simply reflect layer-to-layer propagation of similar representations through the residual stream, rather than revealing a meaningful causal relationship between distinct chunks. This is a valid point: because each layer receives the previous layer's output via residual connections, a chunk extracted at layer L+1 could in principle be identical to the chunk at layer L if the attention module introduces little or no transform.

As a baseline, we computed the cosine similarity between each chunk at layer L and the chunk it causally activates at layer L+1. If causally linked chunks were merely residual copies of one another, their normalized cosine similarity would be close to 1, and the distribution would be sharply centered near 1. This is not what we observe. As shown in the attached one-page figure, the distribution of pairwise similarities between causally connected chunks is broad across layers, rather than concentrated near 1. The spread is especially pronounced in both early input layers and late output layers, suggesting that causal influences in PAG reflect diverse, layer-specific transformations rather than simple duplication. Causally connected chunks are not trivial copies of one another and but distinct representational states.

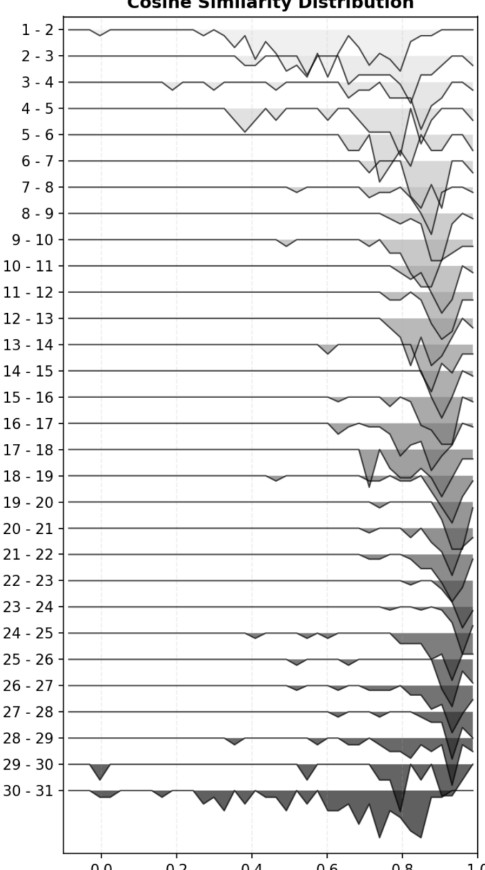

Figure 8: Distribution of normalized cosine similarity between causally connected chunk pairs from the PAG acquired from the *Emma* corpus.

### A.2   DIFFERENCE BETWEEN PAG AND SAE CIRCUITS

We contrast this work with circuit tracing methods and SAEs. SAEs are closely connected to circuit tracing. Before one can trace a circuit, it is necessary to identify which SAE features activate on a given task. These SAE latents then serve as the inputs to circuit tracing: one can determine which attention heads or MLP neurons activate each feature, how those features influence later layers, and how they combine to produce the model's output. The resulting circuit consists of the activated features and the causal edges connecting them, revealing how feature activations propagate across layers. In this way, circuit tracing maps how one feature causally leads to another within a single forward pass of the network. However, there are substantial differences between PAG and circuit tracing methods using SAE.

The first is conceptual. Nodes that constitute circuit in SAE line of work are SAE features, nodes that constitute PAG are chunks. SAE latents contain a continuous activation value in the domain of R+. The activation of one layer in the network is explained by a superposition of SAE latents. The activation of chunks is binary, and one hot, a chunk is there or not. And at one tokenized instance only one chunk from the dictionary explains the observation.

The second difference concerns temporal dynamics. Beyond identifying a computational circuit, PAG reveals how computation unfolds over time—capturing higher-order structure and meaning that develop across tokens. Circuit tracing, in contrast, is mostly agnostic to long-range temporal structure: it extracts a computational graph for a single prompt or forward pass but does not characterize how representations evolve across a sequence. By operating over chunks, PAG instead recovers temporal and hierarchical organization, showing how recurring chunks are sustained, reused, and recombined over the course of a sequence. If circuit tracing is analogous to dissecting the wiring of a single reflex—identifying which neurons connect to which—PAG provides the additional information of when the system transitions from one reflex to another.

The third difference concerns interpretability. Chunks are, by definition, interpretable: they conform to the structure of perceptual understanding, allowing us to recognize recurring patterns as coherent perceptual entities. In contrast, the connection between SAE latents and human-interpretable concepts is often unclear. As a result, circuit tracing typically requires substantial manual inspection—examining many text spans that strongly activate a feature—in order to assign a meaningful label (e.g., a feature that capitalizes the letter "A").

The fourth difference concerns how interventions are performed. Intervening on SAE latents or circuit features typically requires multiplying the feature by a chosen constant, introducing a tunable parameter that must be calibrated. In contrast, activating or inhibiting a chunk requires no such parameter due to their discrete and binary nature. A chunk is there or not. Therefore, chunk-level interventions are also discrete and straightforward, making them much easier to apply and interpret.

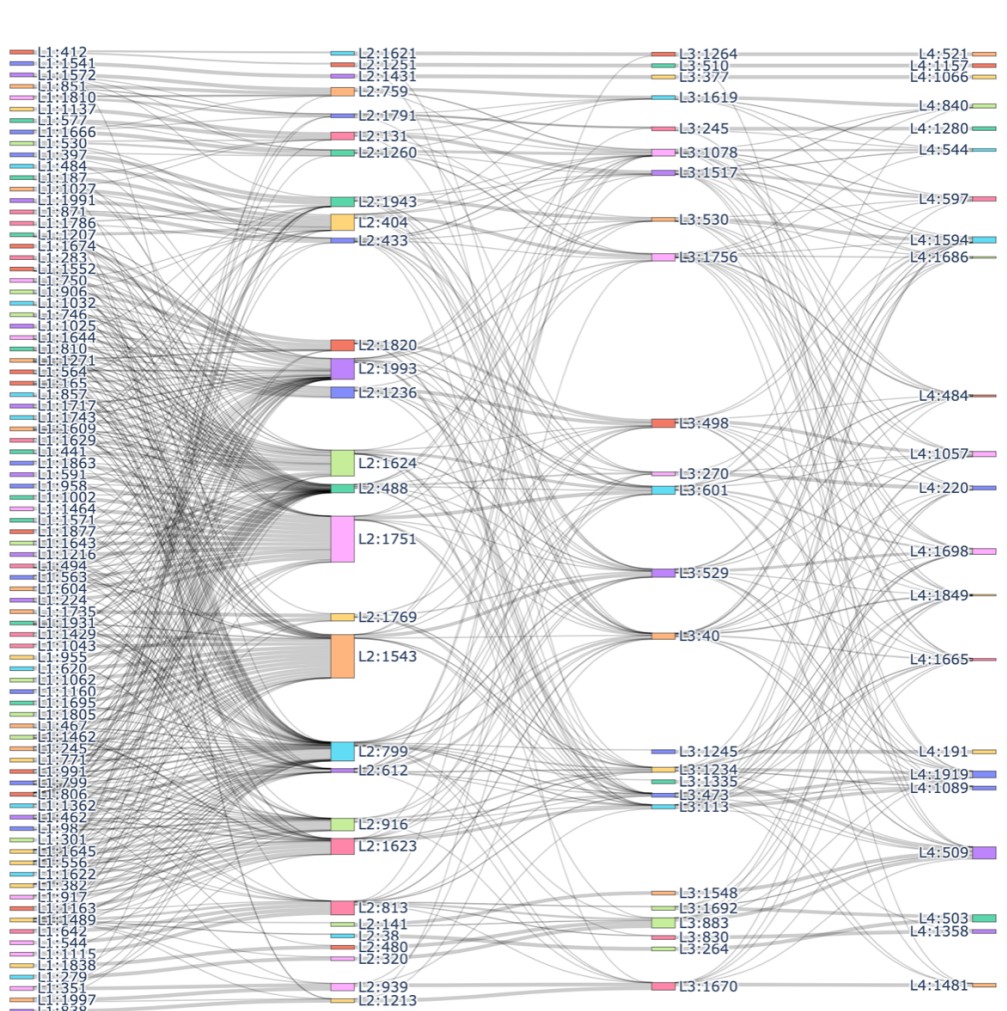

Figure 9: The first 4 layers of PAG extracted from LLaMA-3 processing *Emma* Corpus

