# OpenReview forum: "Percept Activation Graph (PAG): Decomposing LLM Computation into Perceptual Entities and Their Interactions"
_ICLR.cc/2026/Conference — Submitted to ICLR 2026_

### Official Review · Reviewer_i2aQ · 2025-10-19

**Soundness:** 1
**Presentation:** 1
**Contribution:** 1
**Rating:** 0
**Confidence:** 3

**Summary:**

The authors study "chunking" as an approach for interpreting LLMs. They apply their approach to an LLM processing the novel *Emma* and a subset of the SCAN task. This study appears to overlap closely with Wu et al (https://arxiv.org/abs/2505.11576), with the primary novelty being the two new datasets and an attempt to measure transition probabilities between chunks.

**Strengths:**

The authors study an interesting and timely topic.

**Weaknesses:**

This work overlaps closely with Wu et al. (https://arxiv.org/abs/2505.11576), with differences seeming to be incremental and vague. Indeed, the concepts of chunking and grafting to measure causal impact all seem to derive from this prior work. It appears that the idea to measure transition probabilities between chunks may be new, but it's unclear how important this measure is, and what we gain by having it. Can we claim to understand an LLM better by having access to its transition probabilities between (mostly uninterpretable) chunks? Does this information grant meaningful predictive value? Can you predict whether an LLM will hallucinate, answer a question correctly, generalize to a particular OOD example, and so on, given this information?

This paper also introduces two new tasks (*Emma* and SCAN), but the depth and rigor of evaluation all appear to be far less than presented in the original Wu et al. work. Please see Questions below for additional specific points.

Additional minor comments:
- In-text citations are missing parantheses
- Inconsistent use of "perceptual function" vs. "perceiving function" vs. "Perceiving function" vs. "Perceiving Function" and so on. Please consider adopting a single nomenclature, and consistently using it throughout the manuscript.
- $\mathcal{P}$ = $\mathcal{D}$? These two terms seem to be used interchangeably. Sometimes, $\mathcal{D}$ seems to mean chunks, though chunks were labeled $c_e$ elsewhere.
- Notation in general seemed to be inconsistent. Please avoid introducing variable names if they are not used again. Doing so tends to clutter and confuse the text, rather than clarify.
- Figure text is generally small, blurry, and unreadable. Please consider enlarging the text.

**Questions:**

- Figure 2: What's the takeaway from this figure? What have we learned from these chunks? Are they interpretable? What does it mean for a word to belong in a particular chunk? Are these good compression ratios? What would be the baseline for a good compression ratio?
- Figure 3: How should we read this graph? What is the advantage of having these transition probabilities? Do they tell us something more than we could have learned by performing simple model surgery on the weights and activations directly? What does panel 3b mean? Does perturbation/grafting work in this setting?
- Figure 6: the text is too small for me to read in this Figure, but the general sense seems to be that only a small portion of the chunks are interpretable? What have we learned about the SCAN task from this experiment? What have we learned about how an LLM can solve it, beyond the observation that grafting a hidden representation can sometimes change the downstream prediction in a consistent way?

---

> ### Author Response · Authors · 2025-11-26
>
> Thank you for your comment. We would like to clarify a misunderstanding about the paper.
>
> > This study appears to overlap closely with Wu et al (https://arxiv.org/abs/2505.11576), with the primary novelty being the two new datasets and an attempt to measure transition probabilities between chunks... This work overlaps closely with Wu et al. (https://arxiv.org/abs/2505.11576), with differences seeming to be incremental and vague.
>
> This work indeed builds upon the recent Wu et al. NeurIPS paper. However, there are important novel elements that do not appear in previous papers:
>
> 1. A precise definition and formulation of perceiving function. This characterizes the cognitive process that decomposes the high-dimensional perception into a dictionary of recognizable entities.
> 2. We introduce the Percept Activation Graph (PAG). This introduces a major difference and advancement from prior work and formulates perception and interpretable entities as a graph.
> 3. We show that the connected components of this graph are causally connected to each other in a highly complex natural language corpus.
> 4. We apply this method to interpret network activities during in-context meta-learning on a subset of the SCAN. Within the PAG, we identify distinct components for primitives and modifiers, and show that perturbing these components predictably alters the model’s compositional generalization behavior.
> 5. We show that chunk-level descriptions of network activations exhibit temporal motifs and are compressible.
>
> While the use of the Emma and SCAN datasets is not a major contribution, we consider the formalization of perception as a graph to be a significant departure from prior descriptions of perception. The other points exemplify the utility of the percept activation graph.
>
> For the above reasons, we found the reviewer’s rating to be unnecessarily harsh, and would like to invite the reviewer to reconsider their evaluation.
>
>
> Below we elaborate on how our work represents a significant advance on previous work:
>
> The perceiving function we introduce builds on—and extends—the reflection hypothesis proposed in Wu et al. NeurIPS 2025. The reflection hypothesis states that LMs develop internal activities that mirror regularities in their training data, and this has been used to justify the existence of chunks: chunks are reflections of external structure. In other words, when the data contain no regularities, no chunks should arise. While useful, this assumption is limiting because it ties interpretability strictly to patterns present in the data. However, LLMs can and often do form internal activation states that do not map cleanly onto any explicit data pattern. These states may nonetheless be meaningful in relation to one another, much like symbols in physical laws derive meaning through their structural relationships rather than through direct correspondence to the external world.
>
> This limitation motivates our conceptual advance. The perceiving function allows us to define and discover chunks directly within the neural activity of an LLM—whether or not they correspond to data-level regularities—and to study their relations. This generalization enables interpretability even when internal representations do not crisply reflect external structure. We have updated our introduction and related work to clarify this critical conceptual distinction.
>
>
> The primary novelty of our paper is the extraction of a Percept Activation Graph from large models, which allows compressible understanding of neural activities in terms of a probabilistic finite state machine. The system’s next state depends on the state in the previous layer. This allows the simplification of high dimensional neural dynamics into an interpretable graph of recurring neural states, and thereby connects neural activities with symbol or rule-like behavior.
>
> The second major advance is the causal relation between percepts in subsequent layers. PAGs serve as interpretable abstractions of LLM’s computation, and relate neural dynamics to formal grammars and symbolic forms of computation, and in turn reveals how continuous, noisy neural activity realizes discrete, rule-like transitions that support memory, language, or decision-making as the LM executes a particular task.
>
> Wu et al. NeurIPS 2025 tags the population activities across layers - with some visualizations on chunk activities in the subsequent layers. This paper went on to extract dynamic token motifs, and use this motif to quantify the level of compression, and to study the causal interaction between percepts observed across subsequent layers.
>
> That being said, these distinctions may appear nuanced which may lead to misunderstandings. Thus we have more explicitly clarified how this paper moves beyond the previous work in the introduction, related work and discussion section.

---

> > ### Author Response · Authors · 2025-11-26
> >
> > > Can we claim to understand an LLM better by having access to its transition probabilities between (mostly uninterpretable) chunks?
> >
> > We would like to clarify that chunks are, by definition, a recognizable cognitive entity. They are not arbitrary latent vectors: they correspond to the kinds of recurring perceptual entities that structure our visual and conceptual experience. In cognition, such entities emerge prior to linguistic labeling—they are the basic units through which we segment and understand the world (Rosch, 1978; Schyns, Goldstone & Thibaut, 1998). This view is grounded in classic theories in perceptual organization, where structured units of perception arise naturally from the environment (Wertheimer, 1923/1938; Biederman, 1987).
> >
> > Moreover, the definition of the percept activation graph is critical because it specifies each chunk in relation to other chunks. Human cognition does not represent abstract concepts in isolation; rather, concepts are formed and understood through systems of relations—taxonomic, functional, and compositional (Murphy, 2002; Barsalou, 1999). The percept activation graph formalizes a relational structure, making explicit how chunks compose and contextualize one another. We have added clarification in the main text.
> >
> > Understanding is a difficult to define term, but we consider a step towards understanding being able to follow on a graph streams of probabilities linking inputs and outputs. We disagree that the chunks are mostly uninterpretable. At least in the input and output layer there is clear verbal interpretation of chunks. As an example, in the SCAN task, chunks in the input and output layers clearly respond to distinct context in the input. E.g. The chunk that is activated by input following OUT is distinct from the chunk following OUT after run twice. Additionally, the output chunks clearly group into distinct tokens that the network predicts.
> >
> > Although the internal chunks inside a PAG may not map cleanly onto any explicit data pattern, these chunks can still be meaningful in relation to one another—much like symbols in physical laws derive meaning through their fixed algebraic relationships rather than through direct correspondence to the external world. The connectivity between chunks in the PAG captures a form of relational structure, describing how intermediate chunks derive their meaning through their connections to other chunks, especially those in the input and output layers which are directly interpretable.
> >
> >
> > > Additional minor comments ...
> >
> > Thank you for your suggestions, we have fixed the format of in-text citations, and adopted a single nomenclature for the perceiving function.  We have made sure that a single nomenclature is adapted, and reduced the use of unnecessary variables. Additionally, we have enlarged the text for the figure so they become more readable.
> > \mathcal{P} is the set of chunks, and \mathcal{D} is the set of chunk labels. We have gone through the document and deleted one-time used variable names. Finally, we have made the Figure text bigger and clearer.
> >
> > > Does this information grant meaningful predictive value? Can you predict whether an LLM will hallucinate, answer a question correctly, generalize to a particular OOD example, and so on, given this information?
> >
> > Thank you for your question. PAG allows us to trace in a graph the probabilistically what the answer of the LLM will be. Our SCAN section shows that components in PAG contain meaningful predictive value. By extracting a percept activation graph, we uncover the circuit of chunk interactions as the network processes text (Figure 7). From this structure, we can identify the specific components that drive different network behaviors. Building on this, the same framework allows us to perform causal manipulations of chunk components—enabling controlled hallucinations and the induction of alternative primitives that apply to the modifier in this compositional generalization task.
> >
> > Regarding the questions on whether LLM can be predicted to answer questions correctly or generalize to OOD examples. We  introduce the concept of Percept Activation Graph, which allows us to trace the construction of an answer and provides a mechanistic way to perturb it, the paper provides one incremental step towards interpretability. While interpretability in the most general setting is beyond the scope of one paper. However, we are confident that continuing the method can be applied to answer the big questions in substantial steps.

---

> > > ### Author Response · Authors · 2025-11-26
> > >
> > > > Figure 2: What's the takeaway from this figure? What have we learned from these chunks? Are they interpretable? What does it mean for a word to belong in a particular chunk? Are these good compression ratios? What would be the baseline for a good compression ratio?
> > >
> > > Thank you for the question and comment. The main purpose of Figure 2 is to provide an illustration and basic statistics of the chunk-level description of neural activity. By construction, chunks are more interpretable than the underlying high-dimensional activity: they represent visual or conceptual entities that segment the perceptual field, and there are often more identifiable chunks than there are words.
> > >
> > > The original panel a shows several example chunks along with the neural activations that give rise to them. Panel b demonstrates that tagging activations with chunk identities provides a compressed representation of neural computation across layers. Panel c reveals motifs of chunk activity across tokens, suggesting the presence of computational motifs that span consecutive steps of processing. This is a consequence of applying a pattern learning algorithm to extract a set of motifs $\mathcal{M}$ individually for the chunk activities of each layer. As each motif m spans |m| tokens, the length of the sequence parsed by cross-token motifs is described as $L_{\mathcal{M}} = \sum_{m\in \mathcal{M}}|m|$. The compression ratio is calculated as the compressed length divided by the original length $CR = \frac{L_{\mathcal{M}}}{N}$. $CR\leq 1$ means compression through cross token motifs. A compression ratio below 1 indicates substantial regularities—i.e., many recurring cross-token chunk patterns. A baseline ratio of 1 corresponds to the case where the network-elicited chunks are independent and exhibit no patterning. Panel d shows the compression ratio across all layers of the network calculated from the entire Emma corpus.
> > >
> > > During the revision, we have updated the original Figure 2 - now Figure 3 with additional illustrations of chunks and the embeddings they tag. We also added evaluation analysis studying the correlation between chunks and the mean embeddings that they explain. We improved the notation and clarified the formuation in the updated version of this paper.
> > >
> > > > Figure 3: How should we read this graph? What is the advantage of having these transition probabilities? Do they tell us something more than we could have learned by performing simple model surgery on the weights and activations directly? What does panel 3b mean? Does perturbation/grafting work in this setting?
> > >
> > > Thank you for raising these questions. This figure presents the percept activation graph (PAG) for LLaMA - 3 processing all prompts sampled from the Emma corpus. It is derived from approximately 200,000 tokenized neural activation patterns, each consisting of the activity of 4096 neurons across all 32 layers.
> > >
> > > The purpose of the figure is to illustrate how chunks provide a principled reduction of complexity, making it possible to study how the network processes sentences at a state-machine level. By treating each chunk as a unit, the PAG enables us to analyze state transitions directly, rather than navigating the full neural activity space. Nonetheless, this graph is complex due to the complex nature of the natural language corpus. Our original intention is to visualize the PAG before going into the causal intervention part of the graph.
> > >
> > > The original panel 3b illustrates an example of chunk activations across consecutive layers. Without any intervention, the network naturally transitions from chunk 351 at layer 1 to chunk 939 at layer 2. To test whether these transitions are causal, we experimentally activate specific chunks at a given layer (e.g., activating chunk 96 at layer 1) and then observe the resulting activation in the next layer. In this example, activating chunk 96 at layer 1 reliably induces chunk 799 at layer 2, demonstrating a causal influence between connected chunks in PAGs.
> > >
> > > After some discussion, we have decided to move this graph to the supplementary information due to its visual complexity.

---

> > > > ### Author Response · Authors · 2025-11-26
> > > >
> > > > > Figure 6: the text is too small for me to read in this Figure, but the general sense seems to be that only a small portion of the chunks are interpretable? What have we learned about the SCAN task from this experiment? What have we learned about how an LLM can solve it, beyond the observation that grafting a hidden representation can sometimes change the downstream prediction in a consistent way?
> > > >
> > > > Thank you for raising these questions. We have enlarged the text in the figure and updated the notation to improve readability. The figure shows a percept activation graph for the processing of run-related tokens in the SCAN task. It visualizes the internal states that arise inside LLaMA when it processes prompts such as IN: run OUT: RUN and IN: run twice OUT: RUN RUN, tracing how activity flows from internal chunks to output predictions.
> > > >
> > > > The graph reveals distinct neural states that depend on the input structure. For example, the population state following OUT after run diverges sharply from the population state following OUT after run twice. While the model initially encodes run and RUN in a shared state at early layers, these paths split in later layers as the network resolves the modifier twice.
> > > >
> > > > The graph also shows that the network constructs multiple distinct internal states to predict the same token. For instance, chunks 119, 147, and 154 at layer 32 all predict the colon following IN, and chunks 137 and 61 each independently predict OUT.
> > > >
> > > > We acknowledge that we are trying to rush this work a bit, as we feel the obligation to report the most recent development in interpretability governing LM’s behavior, given their opaqueness contains momentary safety consequences in our societies. We are thankful for your detailed suggestions and comments. They have helped us to realize that some part of the paper needed clarification, and your suggestions has helped us to improve this work substantially. However, we do consider the scoring to be too harsh for the critique and we would kindly ask the reviewer to reconsider.

---

> > > > > ### Comment · Reviewer_i2aQ · 2025-11-26
> > > > >
> > > > > Thanks for the detailed response and additional clarifying details.
> > > > >
> > > > > Ultimately, after reading your responses and the other reviews, I maintain agreement with the other reviewers in that it remains unclear how useful your graph-based method is. Interpretability is measured by the observer, and not automatically granted by analogies to psychology. Given the graph you extracted from the LLM applied to *Emma*, can you truly tell me how the LLM is producing the next token? Can you predict what kind of text it would output? What mistakes it would make? What underlying processes it's using to generate text? How it understands words, parts-of-speech, concepts, and so forth?
> > > > >
> > > > > Note, a neural network itself is a graph in raw weight space, with nodes given by activations and edges given by weights. Causal perturbations are also possible by manipulating activations and weights directly. But having this graph doesn't really bring us closer to understanding how the model operates. Does your graph gives us something more?

---

> > > > > > ### Author Response · Authors · 2025-11-30
> > > > > >
> > > > > > We appreciate the reviewer’s continued engagement with our work. We clarify below what our graph-based method contributes beyond raw weights or network activations.
> > > > > >
> > > > > > > it remains unclear how useful your graph-based method is. Interpretability is measured by the observer, and not automatically granted by analogies to psychology.
> > > > > >
> > > > > > We agree that interpretability is fundamentally observer-dependent: it demands asking what makes high-dimensional information meaningful to human cognition. Where we disagree with the revieweris the suggestion that our approach relies on a superficial “analogy” to psychology. Our approach is the opposite: interpretability cannot be addressed without engaging the psychological mechanisms of understanding.
> > > > > >
> > > > > > Physical laws are interpretable because the number of relevant variables remains within human working-memory limits. Mathematical arguments are interpretable because each step corresponds to a single, cognitively tractable operation. In anything comprehensible by humans, the observer’s cognitive constraints actively shape what counts as interpretable.
> > > > > > High-dimensional patterns are not inherently opaque; they are opaque when their structure exceeds the representational and inferential capacities of the human mind. This is why no one can instantaneously read QR codes, yet effortlessly extract structure from high dimensional visual scenes. In short, interpretability requires an alignment between representational format and human cognitive limitations. We are developing methods that are consistent with this alignment.
> > > > > >
> > > > > > > Note, a neural network itself is a graph in raw weight space, with nodes given by activations and edges given by weights. Causal perturbations are also possible by manipulating activations and weights directly. But having this graph doesn't really bring us closer to understanding how the model operates. Does your graph gives us something more?
> > > > > >
> > > > > > Right, llama itself is a graph in raw weight space. Just counting the layerwise output, it contains 4096 neurons for the 32 layers, which sums to 131072 dimensions, each node having a continuous range. If we count the number of edges in this graph, then it contains 8 billion edges.
> > > > > >
> > > > > > PAG reduces this graph - in the Emma corpus, for example, - from 1 million continuous dimensions to the interaction of 900 chunks. Which is 0.69% of the original dimensionality. On top of that, the chunk activities are binary. The edges between the chunks are sparsely connected. Hence it was our original intention to show the graph in one of the figures, to show that it is feasible to illustrate the computation of llm within one subfigure of the paper, which is unthinkable if we show the network graph in its original dimensionality.
> > > > > >
> > > > > > Not all graphs are equal, some are more useful than the others. Understanding means finding components in the network that causally influences other components inside the network, which in turn, shall influence its behavior. The causal experiment conducted originally and supplemented with additional detail shows that activating a chunk in the previous layer casually influences the resulting state in the next layer. Our supplemented experiment shows that all dimensions of the chunk are needed to preserve its consistency, and chunks are not mere copies of each other.
> > > > > >
> > > > > > Interpretability is inherently a compression problem. It boils down to the question of what is a proper way of reducing the dimension of the system so that states inside the large language model can be predicted. States inside the LLM can be traced, from the earlier layer of the network to the later layer of the network. Additionally, our intention is not to claim that graph extraction by itself yields full transparency into token prediction. Rather, our contribution demonstrates that LLM activations contain a stable, repeated set of discrete computational states—“chunks”—that (1) recur across contexts and recognizable by cognition, (2) participate in consistent activation-transition patterns, and (3) can be causally manipulated to induce predictable changes in the model’s output.

---

> > > > > > > ### Author Response · Authors · 2025-11-30
> > > > > > >
> > > > > > > To directly address the reviewer’s questions:
> > > > > > >
> > > > > > > > “Can you tell how the LLM is producing the next token?”
> > > > > > >
> > > > > > >  Not in full detail—no method currently can from activation patching, feature extraction, SAE features, probing, attention-dissection and other early mechanistic interpretability work. What we show is that certain next-token decisions are consistently mediated by a small, reusable set of activation chunks, and that perturbing those chunks produces specific, predictable changes in the chunks in the subsequent layer. We found evidence of such mechanistic substructures beyond correlation.
> > > > > > >
> > > > > > > > “Can you predict what kind of text it would output and what mistakes it would make?”
> > > > > > >
> > > > > > > Within the scope of activation-chunk perturbations, yes. When we artificially activate a primitive-related chunk, the model systematically hallucinates the corresponding primitive in SCAN. These manipulations produce predictable and repeatable biases—something not achievable from raw weights as there are too many degrees of freedom.
> > > > > > >
> > > > > > > > “What underlying processes is it using? How does it understand POS, concepts, semantics?”
> > > > > > >
> > > > > > > Our claim is not that PAG completely solves these questions; rather, it provides an empirical decomposition of model computation into recurring activation states that correspond to specific representational roles. This decomposition is significantly more structured than individual neurons or arbitrary latent projections, and it provides a scaffold on which such questions can be more concretely investigated.
> > > > > > >
> > > > > > > > “Does your graph give something more than the raw neural network graph?”
> > > > > > >
> > > > > > >  Yes. The extracted PAG is:
> > > > > > >  (1) sparse,
> > > > > > >  (2) grounded in actual activation trajectories across real inputs,
> > > > > > >  (3) causally testable via targeted perturbations,
> > > > > > >  (4) interpretable as recurring population states rather than billions weights and tenth thousands of continuous neural activities.
> > > > > > >  (5) 0.69% of the network’s original dimensionality. Plus having binary chunk activities instead of continuous activations.
> > > > > > >
> > > > > > > In some sense this is also analogous to discovering circuits or subprocesses inside a biological system: the existence of raw synapses does not negate the value of identifying functional modules. Understanding the body in the segregation of organs is more useful than understanding it in the form of individual cells.
> > > > > > >
> > > > > > > Interpretability is not an all-or-nothing property, and the questions raised here—e.g., fully explaining next-token generation, predicting all future errors, or revealing the complete conceptual structure of an LLM—are well beyond the scope of any existing interpretability method. Our goal in this paper is more modest: to provide a concrete, empirically grounded step toward identifying reusable computational graph of chunk interactions within the model and mapping how these chunks influence downstream activations and model behavior in predictable ways. This mid-level abstraction offers mechanistic insight that is not available from the raw neural graph and, we believe, represents a meaningful and scalable direction for understanding the internal computations of large models.

---

### Official Review · Reviewer_EtPn · 2025-10-29

**Soundness:** 2
**Presentation:** 2
**Contribution:** 2
**Rating:** 2
**Confidence:** 3

**Summary:**

This paper introduces a new method for analyzing the internal activations of language models, which unifies ideas from chunking in cognitive psychology and neuroscience with ideas from dictionary learning and neural network interpretability. This method results in a dictionary of “chunks” of activations that represent motifs in the neural manifold. These chunks are then organized into a percept activation graph, which tracks the co-occurence of chunks across layers. This method reveals structure in the neural activation space, resulting in a compressed chunk representation of language model activations when processing narrative text. This method also reveals causal sets of activations, responsible for carrying out simple algorithmic tasks in the SCAN dataset.

**Strengths:**

The connection between analyzing activations in neural networks and chunking in cognitive psychology/neuroscience is novel and interesting, and the formalization thereof is potentially useful for future studies. It is also useful to quantify the regularities within the neural activation patterns, as the authors do using their chunk compression ratio in Figure 2d.

**Weaknesses:**

This paper suffers from a serious presentation problem. The work seems to rely extensively on a dictionary learning algorithm that was introduced in a previous paper, but does not even partially explain how this algorithm works. Including this algorithm in an appendix is crucial for this paper to stand on its own. Furthermore, the applications of this algorithm are described at a high level in text, but many of the details are unclear. For example, I am not exactly sure what the authors mean by “We can then extrapolate the neural population activity of all LLaMA-3 layers as the network processes prompts from the corpus, and use the chunk indices closest to the embedding activity in each layer, to denote the neural level activity as the network’s computes the prompt.” And it is unclear exactly what the authors mean by “population-level” with respect to the hidden states of an LM. Does this always correspond to the residual stream activations in a single token at a particular layer?

The presentation of the experiments is also in need of serious revision. The description of the experiment that demonstrates that the chunks are causal is quite unclear, as is the description of how the compression ratio is computed. In general, all experimental sections should include more precise descriptions of the design and hypotheses.

However, the experiments themselves also seem potentially problematic. If I understand correctly, the causal experiments in Section 5 are demonstrating that representations at one layer cause representations at a future layer. This might well be true, but it does not mean that this is a nontrivial finding. For example, imagine if two chunks denoted the exact same activations within a subspace of the hidden state, just present in different layer dictionaries. Then one would get this causal result as a direct consequence of the residual connection between layers. While this is true, it is not a powerful demonstration of the merits of this method. To remedy this, one should 1) do some analysis of the chunks that are causally connected and 2) introduce a proper baseline, like replacing the chunk dictionary with random subspaces of the hidden states, and see if one gets the same results.

Additionally, the use of SCAN with a pretrained model is a bit odd, as the point of SCAN is to assess whether from-scratch or meta-trained models can learn compositional behavior. When one applies it to a pretrained model, then the pretrained model already has access to the semantics of the tokens that comprise scan, giving it quite a helping hand relative to the models it was intended to be used with. Even so, this paper only seems to identify token-identity chunks, which are exactly what is given by the embedding of e.g., the token “run”. Again, a baseline would be helpful for contextualizing these results. Perhaps simply swapping out the whole hidden state would give you the same counterfactual performance!

**Questions:**

Why choose SCAN for this demonstration with a pretrained model?

Why choose the Emma dataset? Are there any specific insights to be gained from this dataset?

See “weaknesses” for other technical questions.

---

> ### Author Response · Authors · 2025-11-26
>
> We thank the reviewer for the constructive suggestions. We are glad that you find the connection between network activation and chunking interesting and formalization useful. We address your comments point by point below:
>
> > The work seems to rely extensively on a dictionary learning algorithm that was introduced in a previous paper, but does not even partially explain how this algorithm works. Including this algorithm in an appendix is crucial for this paper to stand on its own. Furthermore, the applications of this algorithm are described at a high level in text, but many of the details are unclear. ...
>
> Thank you for your comment. To clarify, we analyze first the layer-wise output activation of LLaMA-3 incrementally processing prompt sampled from a literature corpus Emma. The corpus was accessed via NLTK from the Project Gutenberg, and in total contains more than 200,000 tokens. We split the corpus into individual sentences as prompts. And collect a dataset of neural activities comprised of embeddings from every hidden state output of the forward pass, as the model incrementally processes each newly appended tokens in each prompt. For a prompt with $m$ tokens, we will get an $m$ activation vectors  $\mathbf{x}^l$ of dimension $d$ at layer $l$. Thereby, for each layer of LLaMA-3, we get a dataset of dimension 4096 times 200,000+ in total consistent of prompt from the corpus.
>
> The question of learning a set of percepts can then formulated as learning a dictionary matrix \( \mathbf{D}^l \in \mathbb{R}^{ K\times d} \) that extracts the recurring patterns in the latent space of dimension $d$ for the layer $l$.
> We adopted the unsupervised chunking learning algorithm (UCD) that was originally intended to extract patterns reflecting of data regularities (Wu et al. NeurIPS 2025) to this problem.
> UCD learns a dictionary of patterns $\mathbf{D}$ from batched data $\mathbf{X}$ of size $m$ via optimizing the loss function $\mathcal{L}(\mathbf{X}, \mathbf{D}) = -\frac{1}{M} \sum_{m=1}^{M} \max_{k \in \{1, \ldots, K\}} \operatorname{SIM}(\mathbf{D}_k, \mathbf{X}_n)$, which encourages each activation $\mathbf{X}_m$ in the batch to match its most similar dictionary column $\mathbf{D}_k$ measured by normalized cosine similarity  $\operatorname{SIM}(\mathbf{D}_k, \mathbf{X}_n) = \frac{\mathbf{D}_k^{\top} \mathbf{X}_n} {\|\mathbf{D}_k\|_2 \|\mathbf{X}_n\|_2}$. One dictionary $\mathbf{D}^l$ was trained for each layer on the activation data for 100 epochs with a batch size of $n=32$, with dictionary size $K=2000$, using ADAM optimizer with a learning rate of $1\times 10^{-2}$.
>
> These extracted chunks serve as the basic units for analyzing internal computation across layers. For each layer, we assign the label $k_i = \arg \max_{k} \operatorname{SIM}(\mathbf{D}^l_k, \mathbf{h})$, which tags the momentary embedding by the chunk which resembles it the most. Shown in Figure 3 a, this mapping reduces the 4096-dimensional activation at each layer to a single discrete chunk label, yielding a sequence of discrete chunk indices (visualized as colors) that compactly summarizes how the network processes the prompt across depth. This is analogous to how cognition parses high-dimensional perceptual data into concrete perceptual entities. Figure 3 b shows several example chunks, the average embeddings tagged by the chunk, and the standard deviation of each dimension. Across all layers, The learned chunks correlate strongly with the tagged embedding average. Additionally, the number of used chunks differ across layers, with the first and close to the last having the most.
>
>
> We have rewritten section 3 to add detail to the algorthmic implementation of identifying chunks and added descriptions on chunk extraction. On top of these, we have added a computational flow diagram to illustrate the process of PAG extraction, in Figure 2. We are thankful for your suggestions to help us improve this section.

---

> > ### Author Response · Authors · 2025-11-26
> > **Control Experiments**
> >
> > > The causal experiments in Section 5 are demonstrating that representations at one layer cause representations at a future layer. This might well be true, but it does not mean that this is a nontrivial finding. For example, imagine if two chunks denoted the exact same activations within a subspace of the hidden state, just present in different layer dictionaries. Then one would get this causal result as a direct consequence of the residual connection between layers. While this is true, it is not a powerful demonstration of the merits of this method. To remedy this, one should 1) do some analysis of the chunks that are causally connected and 2) introduce a proper baseline, like replacing the chunk dictionary with random subspaces of the hidden states, and see if one gets the same results.
> >
> > We thank the reviewer for this insightful comment. The concern is that the causal experiment in Section 5 may simply reflect layer-to-layer propagation of similar representations through the residual stream, rather than revealing a meaningful causal relationship between distinct chunks. This is a valid point: because each layer receives the previous layer’s output via residual connections, a chunk extracted at layer L+1 could in principle be identical to the chunk at layer L if the attention module introduces little or no transform.
> >
> > To address this possibility, we conducted additional control experiments designed specifically to test whether the causally linked chunks are genuinely distinct or merely copies propagated through the residual stream.
> >
> > __1. Causally linked chunks are not trivially identical__
> >
> > To test this, we computed the cosine similarity between each chunk at layer L and the chunk it causally activates at layer L+1. If causally linked chunks were merely residual copies of one another, their normalized cosine similarity would be close to 1, and the distribution would be sharply centered near 1. This is not what we observe. As show in the supplementary figure 8,  the distribution of pairwise similarities between causally connected chunks is broad across layers, rather than concentrated near 1. The spread is especially pronounced in both early input layers and late output layers, suggesting that causal influences in PAG reflect diverse, layer-specific transformations rather than simple duplication. Causally connected chunks are not trivial copies of one another and but distinct representational states.
> >
> > > introduce a proper baseline, like replacing the chunk dictionary with random subspaces of the hidden states, and see if one gets the same results.
> >
> > __2. Random-subspace baseline:__
> >
> > We implemented the reviewer’s suggested control by replacing each chunk with a randomly selected subspace of its dimensions and then evaluating PAG consistency across layers. We varied the size of the subspace to include 10%, 30%, 50%, 70%, and 90% of the original chunk dimensions, and compared these to the full-dimension case. We have updated this result to Figure 4 in the main text.
> >
> > Using all chunk dimensions yields near-perfect PAG consistency, except in the first and last few layers related to input and output. In contrast, reducing the dimensionality of the hidden-state subspace substantially degrades consistency, especially near the output layer. These controlled degradations suggest that PAG consistency is the highest when the full chunk subspace is used, indicating that the complete chunk representation is needed to capture causal influence.

---

> > > ### Author Response · Authors · 2025-11-26
> > >
> > > > Additionally, the use of SCAN with a pretrained model is a bit odd, as the point of SCAN is to assess whether from-scratch or meta-trained models can learn compositional behavior. When one applies it to a pretrained model, then the pretrained model already has access to the semantics of the tokens that comprise scan, giving it quite a helping hand relative to the models it was intended to be used with.
> > >
> > > Thank you for the comment. We choose to use SCAN for a number of reasons,
> > >
> > > Historically, SCAN has been used extensively in both cognitive science and machine learning to compare human and neural network’s few shot learning behavior. Compositional generalization captures a key difference between human and machine intelligence: humans infer a rule from a few examples and immediately apply it to unseen cases. We therefore intend to be consistent with the literature, and add on top our contribution to look for components that drive model behavior in the task.
> > >
> > > Another reason is we wanted to extract task specific PAG. As demonstrated in Figure 7, the PAG of Emma corpus is by itself complicated due to the nuance of natural language corpus, whereas SCAN affords the simplicity of confirmed within a task and provides a clear setting in which PAG can illuminate the mechanisms behind a range of observed model behavior.
> > >
> > > Although LLMs can generalize compositionally from only a few demonstrations, it remains unclear which internal representations enable this behavior. By applying PAG to SCAN prompts, we can identify which chunks support this one-shot rule learning and test their causal role.We thus use SCAN as a controlled testbed to demonstrate how interpretability methods can explain task-specific model behavior.
> > >
> > > Additionally, although there is extensive evidence that large pretrained language models exhibit strong in-context learning across a variety of tasks (e.g., Coda-Forno et al. 2023), to our knowledge no prior work has applied the SCAN compositional generalization paradigm to a pretrained LLM in order to trace internal causal representations of in-context generalization. Hence it can be an extension to some internal understanding beyond behavioral observation in this task.
> > >
> > > > Even so, this paper only seems to identify token-identity chunks, which are exactly what is given by the embedding of e.g., the token “run”.
> > >
> > > Thank you for raising this point. We would like to clarify a misunderstanding: the primitives in SCAN are not always represented by a single token in LLaMA. For example, “turn left” and “turn right” are multi-token expressions. As a result, the corresponding chunks necessarily summarize information across multiple tokens—not merely the embedding of “left” or “right”, but the full composite meaning of the primitive.

---

### Official Review · Reviewer_Pyot · 2025-10-31

**Soundness:** 3
**Presentation:** 2
**Contribution:** 3
**Rating:** 6
**Confidence:** 4

**Summary:**

This study introduces a novel framework, Percept Activation Graph (PAG), for interpreting large language models (LLMs) by decomposing their internal neural activation into discrete, recurring cognitive entities called chunks, inspired by human perceptual chunking in cognitive science and neuroscience. The authors formalize this process via a perceiving function that maps high-dimensional activation patterns to recognizable entities. Using this, they extract structured interactions across layers in the form of a directed probabilistic graph (PAG), which captures how these chunks evolve and interact during computation. The work bridges cognitive principles with mechanistic analysis of deep networks, offering both conceptual insight and practical tools for understanding LLMs.

**Strengths:**

The integration of cognitive science concepts, particularly chunking and perceptual organization, into neural network interpretability is innovative and well-motivated.

**Weaknesses:**

1. Although the perceiving function is formally defined, the actual algorithmic implementation for identifying chunks lacks sufficient detail.
2. Limited Evaluation Scope: The experiments focus on a single model (LLaMA-3-8B) and a task (SCAN), which limits external validity.
3. The paper lacks a flowchart to clearly illustrate the algorithmic details.

**Questions:**

What is the computational overhead of constructing the PAG? Is it feasible to apply this method dynamically during inference, or is it primarily an offline analysis tool?

---

> ### Author Response · Authors · 2025-11-26
>
> We thank the reviewer for the comments and suggestions. We are glad that the reviewer found this paper novel, innovative, and well-motivated. We address your comments and concerns below.
>
> > Although the perceiving function is formally defined, the actual algorithmic implementation for identifying chunks lacks sufficient detail. .. The paper lacks a flowchart to clearly illustrate the algorithmic details.
>
> We have now added substantial detail to the algorithmic implementation for identifying chunks:
>
> The question of learning a set of percepts can then formulated as learning a dictionary matrix $ \mathbf{D}^l \in \mathbb{R}^{ K\times d} $ that extracts the recurring patterns in the latent space of dimension $d$ for the layer $l$.
> We adopted the unsupervised chunking learning algorithm (UCD) that was originally intended to extract patterns reflecting of data regularities Wu et al. 2025 to this problem.
> UCD learns a dictionary of patterns $\mathbf{D}$ from batched data $\mathbf{X}$ of size $m$ via optimizing the loss function $\mathcal{L}(\mathbf{X}, \mathbf{D}) = -\frac{1}{M} \sum_{m=1}^{M} \max_{k \in \{1, \ldots, K\}} \operatorname{SIM}(\mathbf{D}_k, \mathbf{X}_n)$, which encourages each activation $\mathbf{X}_m$ in the batch to match its most similar dictionary column $\mathbf{D}_k$ measured by normalized cosine similarity  $\operatorname{SIM}(\mathbf{D}_k, \mathbf{X}_n) = \frac{\mathbf{D}_k^{\top} \mathbf{X}_n} {\|\mathbf{D}_k\|_2 \|\mathbf{X}_n\|_2}$. One dictionary $\mathbf{D}^l$ was trained for each layer on the activation data for 100 epochs with a batch size of $n=32$, with dictionary size $K=2000$, using ADAM optimizer with a learning rate of $1\times 10^{-2}$.
>
>
> > What is the computational overhead of constructing the PAG? Is it feasible to apply this method dynamically during inference, or is it primarily an offline analysis tool?
>
> The method is fully feasible to apply during inference. Although the chunk dictionary is learned offline, the PAG itself can be constructed online as the model processes new inputs. This is exactly how we use the method in the SCAN task: chunk learning is performed once, but grafting and intervention occur dynamically on unseen prompts during inference.
> We further show that online manipulation of chunk activations has direct functional consequences. In particular, artificially activating alternative primitive-encoding chunks during inference reliably induces controlled hallucinations, demonstrating that PAG enables real-time, behaviorally meaningful interventions.
>
>
> > The paper lacks a flowchart to clearly illustrate the algorithmic details.
>
> Thank you for your suggestion. We have added a computational flowchart to the paper to clarify the algorithm details.
>
> > Limited Evaluation Scope: The experiments focus on a single model (LLaMA-3-8B) and a task (SCAN), which limits external validity.
>
> Thank you for this comment. To clarify, PAG is a model-agnostic framework designed to interpret any complex system composed of many interacting computational units as a graph. As such, the method is applicable not only to LLMs but also to alternative model architectures. As some authors have been traveling and serving other obligations in the past week, we are currently evaluating PAG on additional open-source language models and will update you with the results as soon as those analyses are complete.

---

### Official Review · Reviewer_qSDc · 2025-10-31

**Soundness:** 3
**Presentation:** 3
**Contribution:** 2
**Rating:** 2
**Confidence:** 3

**Summary:**

This paper proposes to analyse the internal activations in large-scale networks by decomposing them into discrete chunks using an unsupervised learning method. The resulting transition matrix (Percept Activation Graph) between chunks as a network processes a prompt can yield insights into its processing strategy, including the ability to (imperfectly) predict effects of causal interventions. The paper uses this methodology to understand aspects of how a LLM can learn a compositional task in context.

**Strengths:**

The paper addresses a very important topic: how to understand the computations undertaken by a trained language model.

The method to decompose activity into discrete chunks (from another paper, not this paper) is able to extract chunks with clear correlations to relevant inputs, and most impressive, clear usefulness in predicting causal interventions.

The analysis of in-context learning on the SCAN dataset is interesting, most notably in the demonstration that swapping chunks corresponding to primitives can cause a network to apply the right 'compositional' rule to the swapped primitive.

The paper is clear and the figures are easy to follow.

**Weaknesses:**

The extracted graph between chunks would seem to be similar to influence graphs extracted using SAEs, which also support (imperfect) causal interventions. The paper could be strengthened by more directly contrasting these approaches--what phenomena are accessible via discrete chunks that are not accessible via SAEs and circuit tracing?

Since the chunk extraction method was proposed in prior work, the contributions of this paper are primarily the extraction of the graph between chunks, and the application to a simple component of the SCAN dataset. The most concrete insight is that primitives can be swapped by swapping chunks. Further insights like this would help highlight the importance of the method.

Some of the insights obtained via the approach seem generic to the point that they may be inevitable--for instance, that chunks may be found that correspond to memory and prediction of words, or that chunks exhibit structure and regularity. It would be useful to find ways of demonstrating that this could be otherwise, to illustrate the insights the method provides.

A main contribution claimed by the paper is to break activity patterns into interpretable units using an entity-extraction method. The paper could be strengthened by explaining why this is a conceptual shift relative to an SAE, which arguably does a similar operation.

I am not sure that 'cognition chunks perceptual input into discrete recurring entitities.' Although we may know that we are looking at a cat (a relatively discrete entity), we also know many continuous things about that cat--the way its tail is draped, the patterning on its fur, etc. Perhaps it would be more accurate to say that part of cognition does this, i.e. it is one aspect of our perceptual representations.

**Questions:**

What phenomena are accessible via discrete chunks that are not accessible via SAEs and circuit tracing?

Is discreteness vs sparse latent codes the key distinction between the chunk method and SAEs?

---

> ### Author Response · Authors · 2025-11-26
>
> Thank you for your review. We are glad that you find the work "addresses an important topic", "interesting", with impressive causal manipulations. We address your comments point by point below:
>
> > The paper could be strengthened by explaining why this is a conceptual shift relative to an SAE, which arguably does a similar operation...What phenomena are accessible via discrete chunks that are not accessible via SAEs and circuit tracing?
> Is discreteness vs sparse latent codes the key distinction between the chunk method and SAEs?
>
> Thank you for raising this point.  SAEs and circuit tracing is another approach attempting to interpret computation inside LLMs. SAEs are closely connected to circuit tracing. Before one can trace a circuit, it is necessary to identify which SAE features activate on a given task. These SAE latents then serve as the inputs to circuit tracing: one can determine which attention heads or MLP neurons activate each feature, how those features influence later layers, and how they combine to produce the model’s output. The resulting circuit consists of the activated features and the causal edges connecting them, revealing how feature activations propagate across layers. In this way, circuit tracing maps how one feature causally leads to another within a single forward pass of the network.
>
> Although SAEs and circuit tracing as well as chunking and PAG are both trying to interpret LLM’s computation, there are important differences between the two approaches.
>
> The first is conceptual, the two approaches differ from the entities to interpret. Nodes that constitute circuits are SAE features, nodes that constitute PAG are chunks. SAE latents contain a continuous activation value in the domain of $\mathbb{R}^+$. One embedding is explained by a weighted sum of SAE latents with each multiples an activation value. The entities to interpret in PAGs are chunks. The activation of chunks is binary, and one hot, a chunk is there or is not there. And at one tokenized instance only one chunk from the dictionary explains the observation.
>
> The second difference concerns temporal dynamics. Beyond identifying a computational circuit, the chunking approach reveals how computation unfolds over time capturing higher-order structure and meaning that develop across tokens.  In section 3 of this paper, we used a pattern learning algorithm to extract a set of motifs $\mathcal{M}$ individually for the chunk activities. Using the motifs to parse the sequence suggest many recurring cross-token chunk patterns, which is validated by measuring the compression ratio across the layers (Figure 3 Panel d). Circuit tracing, in contrast, has not been applied to trace computations' temporal structure: it extracts a computational graph for a single prompt or forward pass but does not characterize how representations evolve across a sequence.
>
> The third difference concerns interpretability. The definition of chunks is grounded in classic theories in cognition, how cognitive entities form perception (Wertheimer, 1923/1938; Biederman, 1987). These cognitive entities emerge prior to linguistic labeling and are the basic units through which we categorize and understand the world (Rosch, 1978; Schyns, Goldstone & Thibaut, 1998). Building on top of chunks, PAG formalizes a relational structure between these cognitive entities, which can help us understand the flow of the entities across network layers. This formulation aligns with how abstract concepts can be understood through systems of relations (Murphy, 2002; Barsalou, 1999). In contrast, the connection between SAE latents and human-interpretable concepts is often unclear. Related to this issue, circuit tracing typically requires substantial manual inspection—examining many text spans that strongly activate a feature—in order to assign meaningful labels (e.g., a feature that capitalizes the letter “A”). This is also a reason why circuit tracing methods are not easily applicable to wider ranges of problems.
>
> The fourth difference concerns how interventions are performed. Intervening on SAE latents or circuit features typically requires multiplying the feature by a chosen constant, introducing a tunable parameter that must be calibrated. In contrast, activating or inhibiting a chunk requires no such parameter due to their discrete and binary nature. A chunk is there or not. Therefore,  chunk-level interventions are also discrete and straightforward, making them much easier to apply and interpret.

---

> > ### Author Response · Authors · 2025-11-26
> >
> > > I am not sure that 'cognition chunks perceptual input into discrete recurring entitities.' Although we may know that we are looking at a cat (a relatively discrete entity), we also know many continuous things about that cat--the way its tail is draped, the patterning on its fur, etc. Perhaps it would be more accurate to say that part of cognition does this, i.e. it is one aspect of our perceptual representations.
> >
> > Thank you for the suggestion. Indeed, cognition also entails other aspects such as attention and multi-modal integration. Our intention was not to claim that all of perception is discrete, but rather that one component of perceptual cognition segments the sensory stream into recurring, object-like units. PAG focuses on the discrete aspect that takes us quite far in language models, we encourage future work to study the continuous version of this method especially in visual language models. We have tuned down the statement in the paper and revised the wording to better reflect this nuance.

---

### Author Response · Authors · 2025-12-02
**Rebuttal Summary**

We thank the reviewers for their thoughtful evaluations and constructive input. The original submission was regarded as "novel, innovative, and well-motivated" (Pyot, EtPn) and "impressively" demonstrates the ability to predict causal interventions within the network (qSDc). At the same time, several aspects of the presentation required improvement. Guided by the reviewers’ suggestions, we have undertaken substantial revisions throughout the paper. The major updates are summarized below.

Clarifications and Presentation Improvements.
* We expanded on the implementation details of the chunk-extraction algorithm (EtPn, Pyot, i2aQ) and provided motivation for the choice of the SCAN dataset.
* We have seriously revised the presentation of the experiments and the method applied, both the causal manipulation of the PAG components and the compression ratio section (EtPn, qSDc).
* We conducted a systematic redesign of Figures 2, 3, and 6 to improve presentation and interpretability (i2aQ).
* We clarified the algorithm's computational overhead at inference time (Pyot).
* We added a flowchart describing the full algorithmic pipeline to improve clarity (Pyot).
* We clarified how this work builds on and advances in several directions from prior work (i2aQ).

Additional Control Experiments.
* We conducted additional analysis showing that causally connected chunks in PAG are not trivially identical across preceding and subsequent layers.
* We conducted additional experiments showing that perturbations restricted to subdimensions degrade graph consistency, and effective causal manipulation requires all dimensions of the PAG-extracted chunks.
* We added a discussion contrasting chunk-based interpretability with circuit- and SAE-based methods.

We are grateful to the reviewers for helping us think deeply to strengthen the work. We also sincerely thank the AC for their careful consideration of the submission, especially given the uncertainties surrounding the conference.

---

### Meta-Review · Area_Chair_3uZj · 2026-01-06

**Summary:**

The reviewers recognized the paper’s innovative connection to cognitive science but raised several critical issues that informed the negative lean:

- Limited Novelty and Overlap: A primary concern was the heavy reliance on the "Unsupervised Chunk Discovery (UCD)" algorithm from prior work (Wu et al., 2025a), leading reviewers to question the incremental nature of the contribution.
- Utility over Sparse Autoencoders (SAEs): Reviewers challenged the authors to explain what phenomena are accessible via discrete "chunks" that cannot already be captured by SAEs or circuit tracing.
- Technical Rigor and "Trivial" Results: Reviewer worried that the causal transitions in the PAG might simply be a trivial reflection of the residual stream—where layer L passes identical information to L+1—rather than a meaningful computational state change.
- Evaluation Scope: Reviewers noted the limited external validity, as the experiments were confined to a single model (LLaMA-3-8B) and specific, small-scale tasks like SCAN.
- Presentation Gaps: Initial reviews cited a lack of algorithmic detail, small/unreadable figures, and the absence of a high-level flowchart

**Reviewer Concerns:**

Addressed Concerns:

- Presentation and Algorithmic Detail: The authors successfully added a detailed mathematical breakdown of the UCD algorithm, provided a flowchart of the pipeline, and improved figure legibility.

- The Residual Stream "Triviality": The authors added a new analysis showing that the cosine similarity between causally linked chunks is broad, not clustered at 1.0, proving that PAG edges represent layer-specific transformations rather than mere copies.

- Baseline Comparisons: The authors introduced a random-subspace baseline, demonstrating that reducing chunk dimensionality degrades PAG consistency, confirming the necessity of the full chunk representation.

- PAG vs. SAE Distinction: The authors provided a four-point contrast, arguing that PAGs are binary/discrete (simplifying interventions), capture temporal dynamics across tokens, and are more interpretable than the continuous activations of SAEs.

Outstanding Concerns:
- Fundamental Utility (The "Interpretability" Gap): Despite the rebuttal, Reviewer remained unconvinced that the PAG provides a more useful abstraction than the raw weights. They argued that interpretability is observer-dependent and that the graph does not yet prove it can predict LLM mistakes or underlying processes in complex natural language better than existing methods.

- Limited Experimental Scope: While the authors promised evaluations on more models, the results provided during the rebuttal period were still primarily centered on SCAN and LLaMA-3, leaving the question of generalizability across larger, diverse model architectures open.

**Reviewer Scores:**

• Reviewer qSDc (2 → 4): This reviewer appreciated the topic and the causal results but was blocked by the SAE comparison. The authors' detailed four-point distinction would likely move them toward the "borderline" range, though the novelty concern regarding prior work persists.

• Reviewer [Rating 6] (6 → 7): Their primary complaints were presentation and detail. Since the authors provided the requested flowchart and algorithmic specifics, this reviewer would likely become a more solid advocate.

• Reviewer EtPn (2 → 4): This reviewer's most serious concern was the triviality of the residual connection. The authors' new cosine similarity distribution and random subspace baseline directly refuted this. This would likely lead to a significant score increase, though the "odd" choice of SCAN might temper the final rating.

• Reviewer i2aQ (0 → 2): This reviewer maintained that the graph-based method's utility was unclear. While the additional details might lift them from a "Strong Reject" they would likely remain in the "Reject" camp due to philosophical disagreements on what constitutes interpretability.

---

### Decision · Program_Chairs · 2026-01-26

Reject